# OPTIMAL PRICING FOR BUNDLES: USING SUBMODULARITY IN OFFLINE AND ONLINE SETTINGS

## ABSTRACT

We study revenue-maximizing bundle pricing under a cardinality constraint: in each offer the seller chooses a bundle $S \subseteq [n]$ with $|S| \leq k$ and posts a single price $p(S)$. Buyers have unknown valuations $V : 2^{[n]} \to \mathbb{R}$ drawn from a population distribution and purchase with probability given by a choice model that depends on the surplus $V(S) - p(S)$. Based on the seller's ability to collect data on bundles at various price points, we analyze two data regimes:

**Offline.** Given a dataset of past purchases (e.g., receipts) and assuming a logit choice model for customers, we identify near-optimal bundle candidates for bundling. We demonstrate, both theoretically and empirically, that submodularity in the valuation of bundles serves as an effective and sample-efficient criterion for determining promising bundles.

**Online.** With sequential interaction and bandit feedback (sale/no sale) under a more general nonparametric smooth choice model, we design an algorithm with regret of $T^{3/4}$ against an $\alpha$-approximation of the optimal revenue in hindsight, where $\alpha \leq 1 - e^{-1}$ is determined by the supermodular curvature of the expected revenue function.

## 1 INTRODUCTION

Product bundling is ubiquitous in modern marketplaces, appearing in forms as diverse as "buy two get one free" promotions, cable television channel packages, and combo meals at fast-food restaurants. Bundling offers mutual benefits for sellers and buyers by allowing more flexible pricing strategies that can lead to greater market efficiency. However, because the number of potential bundles grows exponentially with the number of products, a central challenge is how to identify promising bundles and determine their optimal pricing—whether in a zero-cost setting (e.g., software suites or online seminars) or when cost is a factor.

In this work, we develop a rigorously justified framework that is both statistically and computationally efficient in identifying bundles within datasets, and that also determines optimal pricing for these bundles in a monopolist setting (i.e., other sellers do not react to our prices). Our approach is based on understanding how the population values combinations of products. By learning this valuation function, we can pinpoint bundles that have the greatest potential to increase overall revenue when sold at appropriately discounted prices.

A key observation in our work is that promising bundles often exhibit a diminishing returns property—that is, the combined value of the products in a bundle is significantly lower than the sum of their individual valuations. This property naturally leads us to the notion of *submodularity*. Informally, submodularity captures the idea that adding an extra product to a smaller bundle increases its value more than adding that same product to a larger bundle. Recognized as a diminishing returns property defined over sets, the following proposition illustrates an important consequence of submodularity on revenue.

**Proposition 1.** *(Informal) Let* $V : 2^{\{1,\ldots,n\}} \to \mathbb{R}$ *be a function assigning scores to any subset of* $\{1,\ldots,n\}$. *If there exists a set of products* $S \subset \{1,\ldots,n\}$, *and two products* $a, b \notin S$ *where* $V(S \cup \{a\}) - V(S) > V(S \cup \{b\}) - V(S)$ *and* $V(a) \approx V(b)$, *then offering* $B := S \cup \{b\}$ *as a bundle at its optimal price leads to a greater increase in revenue than offering* $A := S \cup \{a\}$ *at its optimal price.*

In other words, the set $B$ exhibits stronger diminishing returns—its overall valuation falls short by a larger margin compared to the sum of the valuations of its individual items. In terms of sales, this means that if a customer has already products of set $S$ in a basket, then they are less likely to add product $b$ than product $a$ to the basket. The above proposition suggests that offering the bundle $B$ at a reduced price is an opportunity for increasing revenue. We now formally define submodularity and its close relative supermodularity.

**Definition 1.** *A set function* $f : 2^{\{1,\dots,n\}} \to \mathbb{R}$ *is* **submodular** *if it has the diminishing returns property, i.e. for all* $A \subseteq B \subseteq \{1,\dots,n\}$, *and* $x \notin B$,

$$f(A \cup \{x\}) - f(A) \geq f(B \cup \{a\}) - f(B)$$

*Subsequently, function* $f$ *is* **supermodular** *if for all* $A \subseteq B \subseteq \{1,\dots,n\}$, *and* $x \notin B$,

$$f(A \cup \{x\}) - f(A) \leq f(B \cup \{a\}) - f(B)$$

We will argue that if we can identify bundles with larger "submodularity gap" (the difference between the sum of individual valuations and the bundle valuation), these bundles may be particularly attractive to buyers when priced appropriately. While products with supermodular valuation are more intuitive to bundle (e.g., printer and ink, as they are more valued together than separately), specially if the individual products are no longer available to purchase separately, we show that submodular bundles can be a better choice.

To demonstrate the practical implications of diminishing marginal returns on real data, we first learn a parametric model of the valuation function and compute the revenue impact of adding a bundle using the logit choice model. Empirically, we find that bundles of products with a larger submodularity gap—defined as the difference between the sum of individual product valuations and the valuation of the bundle—result in a more significant increase in revenue. This correlation between the submodularity gap and revenue increase ratio for bundles of two products in a store is illustrated in Figure 1. An intuition for this phenomenon is that items undervalued when sold together may have almost zero demand if priced as the sum of their individual prices. However, offering them as a bundle with a sublinear price can significantly boost their demand and, subsequently, revenue.

With submodularity at the heart of our approach, we focus on two settings for identifying bundles,

- **Offline setting** Given historical data of shopping baskets of customers with fixed prices, the goal is to identify promising bundles of size $\leq k$ that, when priced appropriately, maximize revenue. The idea is that introducing these identified bundles and prices to the store would increase overall revenue over selling just the individual products.

- **Online setting** Here customers arrive sequentially to the seller who offers them a bundle of size $\leq k$ at a specified price. The seller receives the price of the item if the customer purchases it, otherwise the seller receives nothing. The goal is to maximize the total amount of revenue over time. This setting is natural when a bundle displayed as a featured item on a landing webpage, or promoted at the end of an aisle in a retailer store.

In the offline setting, we develop a parametric model built on a logit choice framework that efficiently estimates the average customer valuation for each product. Our approach first leverages historical purchase data to infer product-specific valuations by modeling the probability that a customer chooses a basket of products as a function of their price. These estimated valuations then serve as inputs for predicting both demand and expected revenue when a new bundle is introduced. By integrating a pricing optimization routine, our model identifies the optimal price point for the proposed bundle. To validate our methodology, we apply the revenue estimator to a retailer receipts dataset and empirically examine the relationship between the submodularity gap in the valuation function and the total revenue impact of adding the bundle. This empirical study not only confirms our theoretical predictions but also highlights the practical benefit of capturing submodularity in offering good bundles.

In the online setting, as we can explore the demand curve by offering different price points, we generalize our setting to any nonparametric choice model, and only assume that the environment error of costumer utility is smooth and bounded, i.e. small change to the offered prices would not change the choice probabilities of the customer significantly. A measure of performance over time can be defined as regret against the best possible bundle and price pair in hindsight after $T$ rounds, i.e.

$$\text{Regret}_{T,\alpha} = \alpha \max_{S:|S| \leq k, price \in [0,1]} rev(S, price) - \sum_{t=1}^{T} rev(S_t, price_t),$$

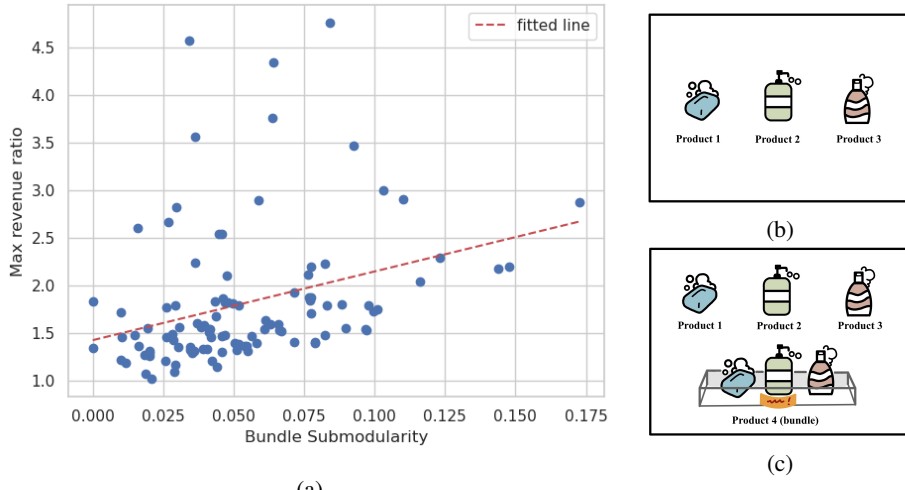

Figure 1: (b) Shop without any bundles, (c) Shop with a bundle offered at the optimal price, with individual product prices unchanged. (a) Plot of the ratio of total revenue, $\frac{rev(\text{Shop c})}{rev(\text{Shop b})}$ for bundles of size 2 versus the submodularity of the bundle's valuation (i.e., the difference between the sum of individual product valuations and the valuation of the bundle) for a subset of product pairs. Bundles exhibiting submodularity tend to provide opportunities for increasing revenue. Our online setting is more general, as even with the assumption of submodularity, there is an additional component of finding the optimal price for each bundle that is being explored.

where $rev(S, price) := price \cdot \text{demand}(S, price)$, is the expected revenue for bundle of $S$ offered with price tag *price*. While in the classic multi-armed bandits literature it is typically the case that $\alpha = 1$, it is known that maximizing a submodular function subject to cardinality constraint requires exponential sample complexity (Vondrák (2013)). Consequently, incorporating $\alpha$ into the regret benchmark reflects the approximation hardness of the revenue function and is frequently found in the submodular optimization literature(Streeter & Golovin (2007); Krause & Golovin (2014)). We propose an algorithm which achieves an $\alpha$-regret of $\widetilde{\mathcal{O}}(T^{3/4})$ assuming the revenue function can be decomposed in a natural way that we will discuss.

## 1.1 RELATED WORK

**Bundle Pricing** Optimal pricing of bundles even for two products is a notoriously hard problem. Most earlier works assume that bundle valuation is additive. Adams & Yellen (1976) showed when there exists a negative correlation in the demand of two products, bundling them is profitable. Lewbel (1985) argued that bundling supplementary products is a better choice. Hanson & Martin (1990) is the first work to address finding optimal bundling algorithmically, for a deterministic costumer with linear valuation. Bakos & Brynjolfsson (1999) and Geng et al. (2005) showed that in a store of information goods where the cost of each product is zero, only offering a bundle of all products(known as pure bundling) under some assumptions can increase the revenue. Manelli & Vincent (2007); Menicucci et al. (2015); Fang et al. (2017); Daskalakis et al. (2017) continued the study of optimal bundling under linear valuation. The literature on nonlinear valuation is limited, and works either focus on only two products(Venkatesh & Kamakura (2003); Armstrong (2013)), or conditions that pure bundling all available products is optimal (Haghpanah & Hartline (2020); Ghili (2022)). In our work we focus on nonlinear valuations for $n$ products, and finding good bundles of constrainted size in both mixed and pure setting.

**Dynamic pricing** Most of the previous work focus on pricing individual products, not subsets. Kleinberg & Leighton (2003) shows $\Theta(T^{2/3})$ regret for adversarial demand pricing of single product, and $\Theta(T^{1/2})$ regret for stochastic demand, making no parametric assumptions. With parametric assumptions on the demand curve family, Broder & Rusmevichientong (2012) shows $\Theta(\log T)$ regret

is possible. Cheung et al. (2017) shows that if the demand curve is from a single parameter family, $\log^{(m)} T$ regret is possible with only $m$ price changes. Maglaras & Meissner (2006); den Boer (2014) study pricing multiple products. There are also works on contextual dynamic pricing, Cohen et al. (2016); Xu & Wang (2021), which can be applied to bundle pricing, however, these works make strictly stronger assumptions analogous to assuming valuation function is additive.

**Submodular Maximization**  Nemhauser & Wolsey (1978); Vondrak (2010) show that monotone submodular maximization under cardinality or matroid constraint is NP-hard and greedy-adjacent algorithms achieve optimal approximation ratio. Greedy algorithm can also give guaranteed approximation for non-submodular functions(Bian et al. (2017); Bai & Bilmes (2018)). For the online environment Streeter & Golovin (2008); Golovin et al. (2014); Nie et al. (2022); Niazadeh et al. (2023); Tajdini et al. (2023) have shown sublinear $\alpha$-regret is possible in various settings. For non-monotone unrestricted maximization $1/2$ approximation ratio is optimal(Feige et al. (2007)), but even under cardinality constraint the optimal ratio is unknown(Buchbinder et al. (2014)).

## 1.2 Contributions

In summary, our contribution in this paper are:

- In the offline setting, we propose a choice model with a quadratic parametrization of the valuation function. First, we show that *submodularity* is a simple and effective measure for identifying promising bundles. Moreover, high-potential submodular bundles can be efficiently identified using greedy procedures. Finally, by modeling the demand and revenue of a shop with added bundles, our framework naturally determines the optimal pricing for each bundle to maximize overall revenue. These claims are supported by rigorous theoretical analysis and validated on a real dataset.
- For the online setting, we propose Algorithm 1 for finding optimal bundle and price pair that maximizes the revenue, and prove that if the revenue function is smooth, and decomposable to submodular and supermodular parts, then Algorithm 1 achieves a $1 - e^{-(1-\beta)}$-regret of $\widetilde{\mathcal{O}}(T^{3/4}n^{1/4}k)$, where $T$ is the time horizon, $n$ is the number of products, and $\beta \in [0,1]$ is an unknown parameter determined by the decomposition of the revenue function.

## 2 Offline setting

In the offline setting we have a dataset of $K$ baskets $\{S_\ell\}_{\ell=1}^K$ where each basket $S_\ell \subset [n] := \{1, \ldots, n\}$ is a subset of all $n$ possible items for purchase. For each item $i \in [n]$ is assumed to have a fixed, known price. To make any inferences about customer behavior, we need a tractable probabilistic model that links our data to the relevant model parameters. Towards this end, this section begins by describing a discrete choice model applicable to an arbitrary valuation function, then defines a tractable parametric valuation model, and then describes how it is efficently learned from data. Finally, we fit the model and demonstrate how to use it to identify promising bundles and predict the relative revenue growth if such a bundle were introduced at the optimal prescribed price.

**Choice Model**  As is common in the pricing literature Benson et al. (2018); Train (2009), we adopt a discrete choice model based on logit distribution over the difference between the customer valuation of a basket and its price. Specifically, given a customer purchased a random bundle $\mathcal{S}$ of exactly $1 \le k \le n$ products, the probability that basket $S \subset [n]$ with $|S| = k$ was purchased is equal to

$$\mathbb{P}\big(\mathcal{S} = S \big| |\mathcal{S}| = k\big) = \frac{\exp(V(S) - price(S))}{\sum_{T:|T|=k} \exp(V(T) - price(T))} \tag{1}$$

where $V : 2^{[n]} \to \mathbb{R}$ is a set function such that $V(S)$ is the expected valuation of a customer to subset $S$ of offered products, and $price(S)$ is the offered price for that subset. Unconditionally, $\mathbb{P}(\mathcal{S} = S) = \mathbb{P}(|\mathcal{S}| = |S|)\mathbb{P}\big(\mathcal{S} = S \big| |\mathcal{S}| = |S|\big)$ where $\mathbb{P}(|\mathcal{S}| = |S|)$ is a discrete distribution over $\mathbb{N}$. By convention, we assume $V(\emptyset) = price(\emptyset) = 0$ so that the probability of buying nothing is equal to $\mathbb{P}(\mathcal{S} = \emptyset) = \mathbb{P}(|\mathcal{S}| = 0)$.

The following lemma formally demonstrates Proposition 1, that submodularity in the valuation function $V$ serves as an effective criterion for identifying optimal bundles.

**Lemma 1.** *(Proposition 1 for $k = 2$) For two pairs of products $\{s, x\}$ and $\{s, y\}$, if $V(\{x\}) = V(\{y\}) = price(x) = price(y)$, and $\frac{V(s)+V(x)}{2} \leq V(\{s, x\}) < V(s, y)$, then with optimal pricing of the bundle constrained to sub-additive pricing ($price(B) \leq \sum_{b \in B} price(b)$), offering the bundle with larger submodularilty gap$(\{s, x\})$ will result in a greater increase in revenue.*

We define optimal price over the sub-additive domain, as offering a bundle where its price is larger than sum of each individual product prices (when individual product are available for purchase) isn't logical, and buyer would buy the products individually. The proof of Lemma 1 is provided in Appendix C.

Consequently, submodularity facilitates the use of efficient approximation algorithms, such as greedy methods, to find bundles that increase the revenue significantly, for large number of products where searching between all bundle choices is infeasible.

**Quadratic model** Assuming the above choice model, our objective is to fit a valuation function $V$. But without additional constraints on the valuation function $V$, it has $2^n$ degrees of freedom making it both computationally and statistically intractable to fit and use for inference. As a consequence, we propose approximating the valuation function with a quadratic model parameterized by a symmetric matrix $A \in \mathbb{R}^{n \times n}$ such that any $S \in [n]$ is valued as

$$\widehat{V}_A(S) = x_S^T A x_S = \sum_{i \in S} A_{i,i} + \sum_{i \neq j \in S} A_{i,j}$$

where $x_S$ is the binary representation of set S in $\{0, 1\}^{[n]}$. Note that our probabilistic choice model equation 2 is invariant to offsets in the values in the sense that the probability under $\widehat{V}_A(S)$ and $\widehat{V}_A(S) + c$ for any $c \in \mathbb{R}$ are equal. Note that this model captures both main and interaction effects(both complimentary and substitutive) of products. As an illustrative, suppose we have a set $S = \{i\}$ and add an element $j$ to it. Now, even if $\widehat{V}_A(\{j\}) > 0$ we could have $\widehat{V}_A(\{i, j\}) \approx \widehat{V}_A(\{i\})$ if $A_{i,j} \ll 0$, a situation that may arise if items $i$ and $j$ are valued highly separately, but serve the same role and are redundant together.

The next claim connects this quadratic model to submodularity.

**Claim 1.** *The set function $\widehat{V}_A : 2^{[n]} \to \mathbb{R}$ is submodular for domain of set $S \subset [n]$ if and only if $A_{i,j} \leq 0$ for all $i \neq j \in S$.*

The degree that a particular pair is submodular/supermodular can be measured by the magnitude of the off-diagonal. Thus, to identify pairs that are submodular and far from modular, we can focus on those with negative interaction terms. Training and evaluation can be found in Appendix B.

**Motivating Example** We consider a shop of just three products: a Maybelline product, a nailcare product, and a Revlon product. Their individual average prices $\bar{p}(\cdot)$ were 2.56, 1.56, and 1.83, and individual valuations of our model of these products where 3.37, 3.43, and 2.99 respectively. If we simply summed these individual valuations, assuming a linear or modular valuation, then the bundle of these three products would be valued at 9.79 but our model predicts the valuation is just 8.54. Intuitively, this suggests that there may be an opportunity to price this bundle lower to drive more sales, and more revenue.

For a bundle $S$, we define the **demand** of $S$ as $\mathbb{P}(S)$ which is just the expected frequency of this basket of products, and the **revenue** as the product of the price and the demand: $p(S)\mathbb{P}(S)$. The prices of all non-bundle will not change so $p(x) = \bar{p}(x)$ for all $x \neq B$. The **total-revenue** of a shop of products is equal to $\mathbb{E}_{S \sim \mathbb{P}}[p(S)]$. One can check that under our model, the demand of $S$ always decreases as $p(S)$ increases. Indeed, in Figure 2 we plot the demand of the individual products in the augmented shop with the additional bundle of the 3 products as a function of the price of the bundle of three products.

We see that as the new bundle is offered with a lower price, the demand of the individual products is lower as customers prefer to just buy the bundle. But as the price of the bundle goes to infinity, the demand for the bundle goes to zero and we converge to the demands enjoyed by the products in the original shop without the bundle.

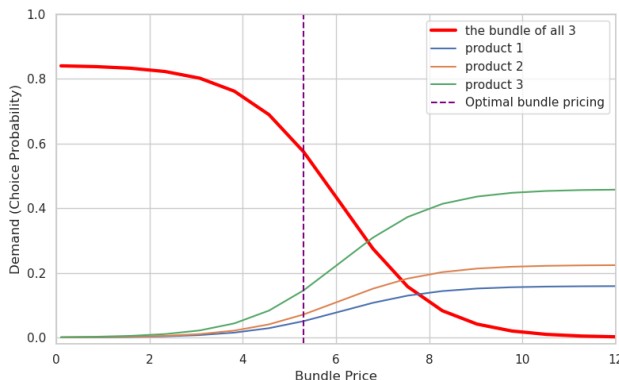

Figure 2: Demands of single products increase, as the price of offered bundle price increases. The dashed line represents the optimal price for the bundle.

By introducing this bundle into the shop we can also study how the total-revenue changes. Note, adding a new bundle to the shop at a lower price than the sum of the individual prices could increase total-revenue by driving more demand at the lower price. However, if the price of the bundle is set too low (in an extreme case, cheaper than the cheapest of three products) then the total-revenue will drop. Figure 3 plots the total-revenue as a function of the price of the added bundle in two scenarios. *Mixed bundling* where the bundle is offered in addition to the individual products, and *Pure bundling* where only the bundle is available for purchase, and individual products cannot be purchased separately. Figure 3b where only the bundle is offered, the revenue is smaller than Figure 3a, where individual products are also offered. In this case, we observe that there is no advantage to creating a bundle and eliminating the individual products for sale (e.g., like how cable providers bundle channels and do not allow a la carte).

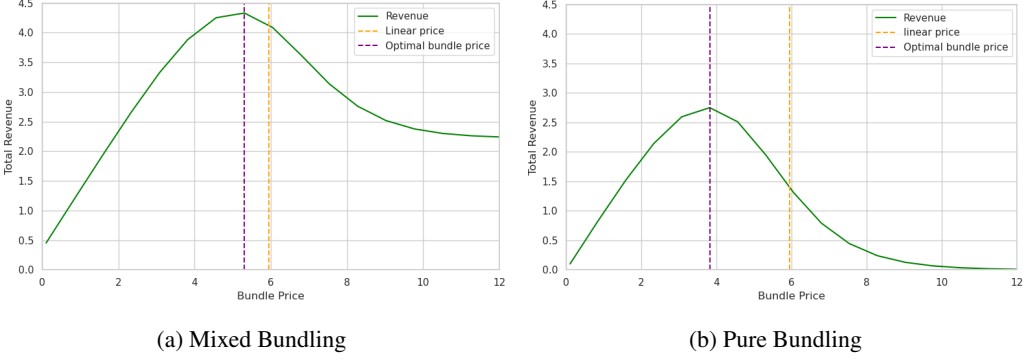

(a) Mixed Bundling                                    (b) Pure Bundling

Figure 3: (a) the bundle of three products available for purchase in addition to all individual products their original price. b) Only the bundle is offered for purchase.

## 3 ONLINE SETTING

The offline setting's analysis relies critically on fitting a parametric model to historical data, which has proven effective for estimating consumer valuations and optimizing bundle pricing in an offline environment. However, this approach is inherently limited. Historical datasets may not capture the full range of possible bundle configurations, especially those that do not occur naturally due to customer self-selection. For example, consumers rarely choose unusually large bundles—such as a package of 60 TV channels—even if such bundles might be profitable under an optimal pricing strategy. This gap highlights a key limitation: if certain bundles are never observed in the offline data, their true demand and revenue potential remain unknown.

To overcome these challenges, we transition to an online setting where we actively collect data by experimenting with bundle offerings at various price points. Importantly, this online framework generalizes to a nonparametric approach, moving away from the rigid structure imposed by parametric models. In this nonparametric setting, we do not assume a specific functional form for the valuation or choice models; instead, we allow the data—collected in real time—to reveal the underlying consumer behavior. This flexibility is crucial, as it enables us to explore and learn from a wider array of bundle configurations, including those that would never naturally occur in historical datasets.

Formally, at each time $t = 1, 2, \ldots$ a buyer arrives with an unknown utility function $U_t : 2^{[n]} \to [0, 1]$ that assigns a non-negative number to any subset $S \subseteq [n]$, where $[n] := \{1, \ldots, n\}$ and $2^{[n]}$ represents all subsets of $[n]$. The utility function can be decomposed $U_t(S) = V(S) + \eta$, where $V$ is the unknown valuation function of the population, and $\eta$ is the mean-zero error parameter of each arriving customer. As the seller, we have no knowledge of the buyer's utility function $U_t$ but offer both a particular bundle $S_t \subseteq [n]$ and post a price $p_t \in [0, 1]$. We assume that if $U_t(S_t) \geq p_t$ then the buyer purchases the bundle $S_t$ at price $p_t$ and the seller receives reward $p_t$. We can succinctly represent our reward or revenue at round $t$ as

$$rev_t := (p_t)\mathbf{1}\{U_t(S_t) \geq p_t\}$$

where $\mathbf{1}\{\cdot\}$ equals 1 if its argument is true, and 0 otherwise. We emphasize that we never observe the buyer's utility function $U_t(\cdot)$ or even $U_t(S_t)$. We only observe whether $U_t(S_t) \geq p_t$ or not. For a given $\alpha > 0$ we define a measure of performance as expected regret:

$$R_T := \alpha \max_{S \subseteq [n], p \in [0,1]} \sum_{t=1}^{T} p\mathbb{P}(U_t(S) \geq p) - \sum_{t=1}^{T} p_t \mathbb{P}(U_t(S_t) \geq p_t).$$

Disregarding $\alpha$, the first term represents the maximum profit achievable in hindsight by choosing a single optimal bundle $S$ and price $p$ for all buyers with utility functions $U_t(\cdot)_{t=1}^{T}$. Even with structural assumptions such as submodularity, there are known sample complexity hardness results for finding the best bundle which are exponential in number of products, so we focus on the widely studies $\alpha$-regret for some $\alpha < 1$ that we'll determine later. We are interested in no-regret algorithms that guarantee convergence to the optimal strategy in the sense that $R(T)/T \to 0$ as $T \to \infty$.

If we consider pricing $K$ individual items (not just a single item) and at each time we offer one of these $K$ items and a price to each successive buyer, then the result of Kleinberg & Leighton (2003) can easily be extended to achieve a regret bound relative to the single best item and price of $O(K^{1/3}T^{2/3})$. Returning to our setting of bundles, if one has $n$ items and a feasible or allowable set of bundles $\mathcal{B} \subset 2^{[n]}$, then we can apply the result with $K = |\mathcal{B}|$. However, since $|\mathcal{B}|$ could be as large as $2^n$, this translates into a regret bound scaling like $2^{n/3}T^{2/3}$. If the number of items $K$ is large, this amount of regret is unacceptably large, and moreover, this implies that the performance of any algorithm is vacuous until $T \geq 2^{n/3}$, an unacceptably long time.

Fortunately, with minor assumptions about the expected revenue function $rev(p, S)$ for each bundle $S$ offered with price $p$, the search problem can become much simpler. In particular, *monotonicity* states that for any two subsets $A \subseteq B \subseteq [n]$ we have $max_p rev(p, A) \leq max_p rev(p, A)$. This assumption is intuitive: since bundle $B$ contains all the items of bundle $A$, it is at least as desirable as $A$. Offering both bundles at the same price will result in higher demand for the larger bundle. Therefore, with optimal pricing, the larger bundle will generate at least as much revenue as the smaller one. Secondly, the revenue function at optimal price has *SuBmodular+SuPermodular(BP) decomposition* decomposition into sum of non-decreasing submodular and supermodular components (recall the definitions introduced in the introduction). This assumption is also intuitive as the products in a basket can have complimentary and substitutive interactions, and the effects of these interaction on the revenue can subsequently be decomposed.

**Assumption 1.** *Let $rev(p, S) = p\mathbb{P}[U(S) \geq p]$ be the expected revenue function for a price $p$, and bundle $S$, then assume*

1. *$rev$ has a unique maximizer $p^*$ in $(0, 1]$ and is 1-Lipschitz i.e. for all $p, p' \in [0, 1]$, $rev(p, S) - rev(p', S) \leq |p - p'|$,*

2. *$rev^*(S) := \max_p rev(p, S)$ can be decomposed as $f(S) + g(S)$, where $f$ is submodular, $g$ is supermodular, and both are non-decreasing.*

Assumption 1 offers a more relaxed perspective on population behavior while preserving the computational efficiency in identifying optimal bundles that submodularity provides. This balance allows for a broader application of our model without compromising the tractability of the solution process.

**Remark 1.** *If the demand distribution $\mathbb{P}[U(S) \geq p]$ is the exponential distribution with $\lambda = \frac{1}{V(S)}$, then the revenue is maximized at $p = V(S)$, and we have $rev^*(S) = e^{-1}V(S)$. Therefore, the existence of a submodular+supermodular decomposition for the valuation function directly implies that the revenue function enjoys the same decomposition, and in particular, Assumption 1 holds.*

**Remark 2.** *The demand distribution $\mathbb{P}[U(S) \geq p]$ is continuous if and only if $rev(p, S)$ is 1-Lipschitz.*

---

**Algorithm 1** Deterministic Greedy Dynamic Pricing

---

**Input:** $T, k, n, m, M$ price discretization
**Initialization:** $\bar{P} = \{\frac{1}{M}, \frac{2}{M}, \ldots, \frac{M}{M}\}$, and $S^{(0)} = \emptyset$
**for** $i = 1, 2, \ldots, k$ **do**
    **for** $a \notin S^{(i-1)}, \bar{p} \in \bar{P}$ **do**
        **for** $j = 1, \ldots, m$ **do**
            Play $S_a := S^{(i-1)} \cup a$ with price $\bar{p}$, receive reward $r_t = \bar{p}\mathbf{1}(U_t(S_a) \geq \bar{p})$.
            $t \leftarrow t + 1$
        Compute $\widehat{\mu}_{S_a,\bar{p}} = \frac{1}{m}\sum_{t:(S_a,\bar{p})\text{ played}} r_t$ for all $a \notin S^{(i-1)}, \bar{p} \in \bar{P}$
    Update the base set: $S^{(i)} \leftarrow S^{(i-1)} \cup \{a_i\}$ where $a_i := \arg\max_a \max_{\bar{p} \in \bar{P}} \widehat{\mu}_{S_a,\bar{p}}$
    Store the selected price: $p^{(i)} \leftarrow \arg\max_{\bar{p} \in \bar{P}} \widehat{\mu}_{S^{(i)},\bar{p}}$
**while** $t < T$ **do**
    Play $(S^{(k)}, p^{(k)})$ and update $t \leftarrow t + 1$

---

Algorithm 1 is an explore-then-commit algorithm, where it builds a bundle by adding elements greedily after enough exploration at each round. To find the best element to add to the base set at each cardinality, it discretizes the price choices.

**Greedy Approach:** To maximize the set function $rev$ under a cardinality constraint $k$, we employ a greedy approach. Starting with the empty set, the algorithm iteratively adds elements that provide the highest marginal gain in $rev$, continuing this process until the set reaches size $k$. At each iteration, an error margin of $\epsilon$ is tolerated, allowing for near-optimal selections that balance sample complexity with solution quality.

**Discretization and Pricing Optimization:** Given that the function $rev$ incorporates a pricing component which is Lipschitz, we discretize the price range into intervals separated by $\epsilon$. This discretization enables the application of multi-armed bandit algorithms to identify the optimal price within an $\epsilon$-precision.

Consequently, each step of the algorithm introduces an error of up to $2\epsilon$, accounting for both the selection process and the discretization of prices.

## 3.1 REGRET ANALYSIS

**Theorem 1.** *If Assumptions 1 and 2 hold, then algorithm 1 with $m = T^{1/2}/\log^{1/2} T$ and $M = T^{1/4}/\log^{1/4} T$ achieves $1 - e^{-(1-\kappa_g)}$-regret bounded by*

$$(2 + 2\sqrt{2})T^{3/4}n^{1/4}k\log^{2/3}(2knT^3),$$

*where $\kappa_g \in [0, 1]$ is the supermodular curvature(see Bai & Bilmes (2018)) of function $g$ in the BP decomposition.*

In order to prove the regret bound, we show that errors in the greedy algorithm are additive, i.e. if we have an error of $\epsilon$ at each step, then the solution has only a gap of at most $k\epsilon$ with the guaranteed approximation ratio.

**Lemma 2.** *(Generalization of Lemma 3.6 in Bai & Bilmes (2018) with additive error) Let $S_0 \subset S_1 \subset \ldots \subset S_k$ be an $\epsilon$-greedy chain for BP function $h$, then*

$$h(S_k) + k\epsilon \geq (1 - e^{-(1-\kappa_g)})h(S^*)$$

We provide a proof in Appendix D. When the function is submodular and exhibits no supermodular behavior (i.e., $\kappa_g = 0$), then Lemma 2 becomes equivalent to Theorem 6 in Streeter & Golovin (2007).

So far we made no assumptions on the demand curve; however, it is possible to achieve better regret bound with Algorithm 1 if the demand curve is concave.

**Assumption 2.** *If $rev(p, S) = p\mathbb{P}[U(S) \geq p]$ is the expected revenue function for a price $p$, and bundle $S$, then $\frac{\partial^2}{\partial p^2} rev(p, S) < 0$ for $p \in (0, 1)$.*

**Theorem 2.** *(Informal) If Assumption 2 holds for all subsets $S$ where $|S| \leq k$, then the regret of Algorithm 1 is $\widetilde{\mathcal{O}}(T^{5/7})$.*

The proof can be found in Appendix E.

## 4 CONCLUSION

In this work, we have introduced a rigorous framework for both identifying promising product bundles and determining their optimal pricing in two distinct settings—offline and online. By leveraging historical purchase data and a logit choice model, our approach estimates individual product valuations and exploits the submodularity (i.e., diminishing returns) property inherent in some bundled offerings. Empirical results confirm that bundles with larger submodularity gaps can significantly boost revenue, thereby validating both our theoretical predictions and our computational methods.

Looking ahead, several promising directions can extend and enrich our work.

**Maximizing Profit Instead of Revenue:** While our current framework focuses on revenue optimization, a natural extension is to incorporate the cost of items to directly maximize profit. While for the parameterized offline setting we can directly compute the profit with the fitted valuation function, this shift introduces significant challenges for building an optimal bundle greedily since the profit function may not be monotonic with respect to the set function of products. For the special case of submodular valuation over all products, we propose a Algorithm 2, a randomized greedy algorithm that can achieve sublinear $e^{-1}$-regret. Exploring methods to generalize our approach without restricting the profit function class to submodular, and allowing supermodular components, presents a promising direction for future research.

**Refined Valuation and Choice Models:** Our study of the offline setting currently employs a parametric model—based on a relatively simple quadratic approximations to estimate customer valuations. Future work could explore more complex and expressive models that capture richer consumer preferences. In addition, extending the analysis to incorporate more general choice models, potentially addressing heterogeneous customer behavior and nonparametric frameworks, may further enhance the robustness and applicability of the proposed methods.

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

# A  APPENDIX

# B  TRAINING AND EVALUATION

## B.1  TRAINING

Given our discrete choice model linking a valuation function to observed baskets and our quadratic valuation model, fitting the model to a dataset is as easy as computing the maximum likelihood estimate for $A \in \mathbb{R}^{n \times n}$. However, when implementing a gradient descent procedure one quickly realizes that the necessary computation of the denominator of equation 1 is a sum over $\binom{n}{k}$ terms–an intractable computation for large $n$. Instead of computing it directly, we aim to estimate it using importance sampling with a convenient candidate distribution $Q$ that is the result of a "pre-training" step. Specifically, we use the unbiased estimator

$$Z := \sum_{|S|=k} \exp(\widehat{V}_A(S) - \mathrm{price}(S)) \approx \frac{1}{T} \sum_{t=1}^{T} \exp(\widehat{V}_A(S_t) - \mathrm{price}(S_t))/Q(S_t)$$

where $S_t \sim Q(\cdot)$ drawn IID. For our choice of candidate distribution $Q$ defined below, we demonstrate in Appendix G that this estimator converges in just several hundred samples. We then simply use the above plug-in estimate for the denominator in our gradient descent procedure, recomputing this estimate each iteration with the current iterate of the valuation function.

**Candidate Distribution Pretraining**  Our objective is to define a candidate distribution $Q$ over bundles that is cheap to sample from that is "close enough" to the true distribution to reduce the variance of the resulting estimator. Towards this end, we take inspiration from Ruiz et al. (2019) and adopt an auto-regressive model. Specifically, we model a basket of items $S \subset [n]$ as being built up one product at a time so that the third product added is drawn from a distribution conditioned on the first two products added to the set. At each time we also include the potential for a STOP token to be drawn which terminates the basket. Thus, for a basket $S = \{x_1, \ldots, x_l\}$

$$\mathbb{P}(S) = \mathbb{P}(x_1|\emptyset)\mathbb{P}(x_2|\{x_1\})\mathbb{P}(x_3|\{x_1, x_2\}) \cdots \mathbb{P}(\mathrm{STOP}|S).$$

We assume each conditional distribution is a logit model where the conditional valuation of the union set of that product and previous product is considered relative to the combined price:

$$\mathbb{P}(x|S) = \frac{\exp(V(x \cup S) - price(x \cup S))}{\sum_{y \in [n] \cup \{\mathrm{STOP}\} \setminus S} \exp(V(y \cup S) - price(y \cup S))} \tag{2}$$

for all $x \in ([n] - S) \cup \{\mathrm{STOP}\}$. Such a model is straightforward to fit to a dataset given our parametric valuation model using gradient descent. And most importantly for our needs, it is very efficient to sample baskets from as we will use it as our candidate distribution $Q$.

A natural question is: why not use the autoregressive model as our primary model for inference? The main issue is that this model introduces an artificial temporal component that does not exist

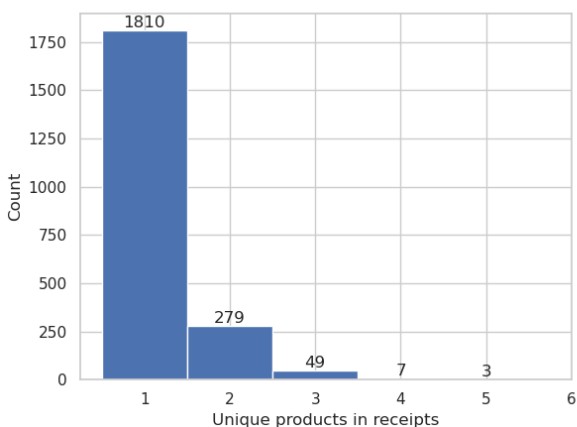

Figure 4: Histogram of unique products in receipts

in the data. In other words, adding products in different orders leads to different probabilities for the same set. To address this, Ruiz et al. (2019) proposed drawing a random permutation in each training iteration so that the model effectively places a uniform measure on all orderings. However, this approach means that inference now requires averaging over many permutations to calculate probabilities. A second reason for our choice is simplicity. The logit discrete model is well studied and is the default choice in the literature for customer decision modeling. Moreover, our model is much simpler and makes it easier to understand how a submodular valuation function translates into probabilistic behavior—an insight that will be valuable when considering an online model. Finally, the sequential model doesn't capture the effects of the size of the basket in the choice model, and is based on only the average compatibility of a new product with products already in the basket.

## B.2 MODEL INFERENCE

Given our trained model, we now describe how one can make inferences about how adding a bundle at a specific price would impact overall revenue given that the shop of potential purchases has changed. For example, if the original shop of products was $\{a, b, c\}$ and we identified a bundle of size 3 $\{a, b, c\}$, the new shop of available products to purchase would be $\{a, b, c, d\}$ where $d = \{a, b, c\}$ is the bundle of products. (Figures 1b-1c).

As we have fitted the set function $\widehat{V}_A$, we can evaluate this function to *any* set we'd like and ask the counterfactual question of what would be the demand of each product(including the bundle) under the new shop of products with the additional bundled set with any price that we desire. Specifically, if a bundle $B \subseteq [n]$ is added to a shop of $n$ products, and the valuation function of the new shop would be

$$\widehat{V}'(S) = \widehat{V}_A(\{x \in [n] | x \in S \text{ or } x \in B \in S\}) \qquad \forall S \subseteq [n] \cup \{B\},$$

as we are assuming that duplicates of a product is not possible. Therefore, the basket choices are $\mathcal{C} = \{S \subseteq [n] \cup \{B\} | B \notin S \text{ or } S \cap B = \emptyset\}$, the expected revenue of the new shop would be

$$\sum_{S \subseteq \mathcal{C}} \text{price}(S) \mathbb{P}(\mathcal{S} = S) = \sum_{S \subseteq \mathcal{C}} \text{price}(S) \sum_{k=0}^{n+1} \mathbb{P}(\mathcal{S} = k) \frac{\exp\left(\widehat{V}'(S) - \text{price}(S)\right)}{\sum_{T \subset \mathcal{C}, |T| = k} \exp\left(\widehat{V}'(T) - \text{price}(T)\right)}.$$

## B.3 EXAMINING RETAILER DATASET

We use the dataset of a retailer store purchases over two years[1] from over 2500 households, with purchase receipt of bought products and their prices for every shopping trip they had. To aid

---

[1]"The Complete Journey", https://www.dunnhumby.com/source-files

interpretability of identified bundles and reduce the ground set, we limited our study to beauty products. Within this reduced set of receipts, there are a total of 2148 receipts and 282 total products, with an average of 1.19 unique products purchased per receipt (see Figure 4 for a histogram of number of unique products per receipt). The $i$th receipt is a list of products $S_i$ and prices $\{p_i(j)\}_{j \in S}$, and we fit the value function model $\widehat{V}$ using maximum likelihood estimation of the choice model with ridge parameter of 0.01 over off-diagonal entries of $A$. However, we first normalize the prices across time for simplicity, and define $\bar{p}(j) = \frac{1}{N_j} \sum_{i:j \in S_i} p_i(j)$ where $N_j$ is the number of receipts product $j$ appeared on. If the same product is purchased more than once on a single receipt, we ignore the cardinality and act as if just a single purchase of the product was made. As all the prices of a dataset without bundles are on individual products, we have $\bar{p}(S) = \sum_{x \in S} \bar{p}(x)$. Using 10 percent of data as validation set, $\lambda = 0.01$ is the optimal ridge term, and we use SGD with learning rate 0.05 as the optimizer.

Given this offline dataset, our goal is to identify a set of products that, if bundled and repriced, can lead to increased revenue. We expect sets of products that have a high degree of submodularity to be precisely those that exhibit the ability to drive revenue when bundled.

## C    Proof of Lemma Lemma 1

We assume $V$ includes the set cardinality probabilities w.l.o.g., as the choice model is invariant to additive constants of each cardinality, and logit choice models is between all possible subsets. The expected revenue when offering bundle for $\{s, x\}$ with price $p_B$ is

$$\frac{p_1}{\exp(V(\emptyset)) + \exp(V(\{s\}) - p_1) + \exp(V(\{x\}) - p_2) + \exp(V(\{s, x\}) - p_B)}$$

$$+ \frac{p_2}{3 + \exp(V(\{s, x\}) - p_B)} + \frac{p_B \exp(V(\{s, x\}) - p_B)}{3 + \exp(V(\{s, x\}) - p_B)},$$

where $p_1 = \text{price}(s)$, and $p_2 = \text{price}(x) = \text{price}(y)$. Similarly, the original expected revenue without the bundle is

$$\frac{p_1 + p_2}{3 + \exp(V(\{s, x\}) - (p_1 + p_2))} + \frac{(p_1 + p_2) \exp(V(\{s, x\}) - (p_1 + p_2))}{3 + \exp(V(\{s, x\}) - (p_1 + p_2))},$$

.

Therefore, the optimal price $p'_x = \min(p^*_x, p_1 + p_2)$ for the offered bundle of $\{s, x\}$, where the stationary point $p^*_x$ is the solution of $V(\{s, x\}) = p^*_x + \log(3p^*_x - p_1 - p_2 - 3)$. So, the revenue under optimal pricing would be

$$\left( \frac{p_1 + p_2}{3 + (3p^*_x - p_1 - p_2 - 3)} + \frac{p^*_x(3p^*_x - p_1 - p_2 - 3)}{3 + (3p^*_x - p_1 - p_2 - 3)} \right)$$

$$- \left( \frac{p_1 + p_2}{3 + (2p^*_x - p_1 - p_2 - 3)\exp(p_1 + p_2 - p^*_x)} + \frac{(p_1 + p_2)(3p^*_x - p_1 - p_2 - 3)}{3 + (3p^*_x - p_1 - p_2 - 3)\exp(p_1 + p_2 - p^*_x)} \right)$$

This function is decreasing for $\frac{p_1 + p_2}{2} + 1 \leq p^*_x \leq p_1 + p_2$. Similarly, the optimal price $p'_x = \min(p^*_y, p_1 + p_2)$ for the offered bundle of $\{s, y\}$ where $p^*_y$ is the solution of $V(\{s, y\}) = p^*_y + \log(3p^*_y - p_1 - p_2 - 3)$. Concluding that since $V(\{s, x\}) < V(\{s, y\})$, then $p'_x < p'_y$, and the total increase in revenue when offering bundle of $\{s, x\}$ is higher.

## D    Additive error for general supermodular curvature $\kappa_g$

**Lemma 3.** *(Lemma 3.5 in Bai & Bilmes (2018)) for any nested subset chain $S_0 \subset S_1 \subset \ldots \subset S_k$, where $|S_i| = i$, and monotone BP function $h$, we have*

$$h(S^*) \leq \kappa_f \sum_{j:s_j \in S_i \setminus S^*} h(s_j | S_{j-1}) + \sum_{j:s_j \in S_i \cap S^*} h(s_j | S_{j-1}) + h(S^* \setminus S_i | S_i)$$

We provide the bounds for worst-case submodular curvature of $f$ and only parametrize supermodular curvature of $g$ by $\kappa_g$.

*Proof.* Since the chain is $\epsilon$-greedy, for all $v \notin S_i$, $h(s_{i+1}|S_i) + \epsilon \geq h(v|S_i)$, and using Lemma 3,

$$h(S^*) \leq \kappa_f \sum_{j:s_j \in S_i \setminus S^*} h(s_j|S_{j-1}) + \sum_{j:s_j \in S_i \cap S^*} h(s_j|S_{j-1}) + \frac{k - |S^* \cap S_i|}{1 - \kappa_g} h(s_{i+1}|S_i) + \frac{k - |S^* \cap S_i|}{1 - \kappa_g}\epsilon$$

$$\leq \sum_{j:s_j \in S_i \setminus S^*} h(s_j|S_{j-1}) + \sum_{j:s_j \in S_i \cap S^*} h(s_j|S_{j-1}) + \frac{k - |S^* \cap S_i|}{1 - \kappa_g} h(s_{i+1}|S_i) + \frac{k\epsilon}{1 - \kappa_g}\epsilon$$

Therefore, following proof of Lemma D.2 from Bai & Bilmes (2018), for nonnegative variables $a_1 \ldots, a_k$, we define the following LPs for each subset of indices $B \subset [k]$:

$$T(B) = T(b_1, b_2, \ldots, b_p) = \min_{a_1, a_2, \ldots, a_k} \sum_{i=1}^{k} a_i$$

subject to

$$h(S^*) - \frac{k\epsilon}{1 - \kappa_g} \leq \alpha \sum_{j \in [i-1] \setminus B_{i-1}} a_j + \sum_{j \in B_{i-1}} a_j + \frac{k - |B_{i-1}|}{1 - \beta} a_i, \text{ for } i = 1, \ldots, k.$$

As $h(S^*) - \frac{k\epsilon}{1 - \kappa_g}$ is a constant in all of LPs, we can replace it with $h'(S^*) := h(S^*) - \frac{k\epsilon}{1 - \kappa_g}$, and it is shown that $T(\emptyset) \leq T(B)$ for all $B \subset [k]$.

We fix $\alpha = 1$, and $\beta = \kappa_g$, then for any feasible solution $(a_1 \ldots, a_k)$ of $T(\emptyset)$, defining $T_i = \sum_{j=1}^{i} a_j$, we have

$$h(S^*) - \frac{k\epsilon}{1 - \beta} \leq T_{i-1} + \frac{k}{1 - \beta}(T_i - T_{i-1})$$

, so by multiplying by $\frac{1-\beta}{k}$, we have

$$h(S^*) - T_i \leq \left(1 - \frac{(1-\beta)}{k}\right)(h(S^*) - T_{i-1}) + \epsilon$$

and repeatedly applying all $k$ inequalities,

$$T_k \geq \left[1 - \left(1 - \frac{(1-\beta)}{k}\right)^k\right] h(S^*) - k\epsilon$$

As $a_i := h(s_i|S_{i-1})$ for $\epsilon$-greedy chain $S_0 \subset S_1 \subset \ldots \subset S_k$, is a feasible point in LP of $B$ for $B = S_k \cap S^*$, then

$$h(S_k) \geq T(B) \geq T(\emptyset) \geq \left[1 - \left(1 - \frac{(1-\beta)}{k}\right)^k\right] h(S^*) - k\epsilon$$

$\square$

## E CONCAVE DEMAND CURVE REGRET

**Lemma 4** (Lemma 3.11 in Kleinberg & Leighton (2003)). *For any demand curve satisfying Assumption Assumption 2 there exists constants $c_1, c_2$ such that $c_1(p - p^*)^2 \leq f(p^*) - f(p) \leq c_2(p - p^*)^2$*

**Theorem 3.** *If Assumption 2 also holds in addition to Assumption 1 and 2, then with high probability then algorithm 1 with $m = T^{4/7}/\log^{4/7} T$ and $M = T^{1/7}/\log^{1/7} T$ achieves $R_{gr}$-regret $\mathcal{O}(T^{5/7})$.*

*Proof.* Similar to the proof of Theorem 1 on event $G$, we have

$$f(S^{(i-1)} \cup \{\bar{a}^*\}, \bar{p}^*) - f(S^{(i)}, p^{(i)}) \le 2\sqrt{\frac{2\log(2knMT^2)}{m}}$$

Now let $a^* = \arg\max_a f^*(S^{(i-1)} \cup \{a\})$, where $f^*(S) = \max_{p \in [0,1]} f(S, p)$, and $p^* = \arg\max_{p \in [0,1]} f(S^{(i-1)} \cup a^*, p)$. $\frac{\lfloor Mp^* \rfloor}{M}$ is in discretized price set $\bar{P}$, and $p^* - \frac{\lfloor Mp^* \rfloor}{M} \le \frac{1}{M}$. Therefore, using Lemma Lemma 4,

$$f(S^{(i-1)} \cup \{a^*\}, p^*) - f(S^{(i-1)} \cup \{\bar{a}^*\}, \bar{p}^*)$$
$$\le f(S^{(i-1)} \cup \{a^*\}, p^*) - f(S^{(i-1)} \cup \{a^*\}, \frac{\lfloor Mp^* \rfloor}{M})$$
$$+ \underbrace{f(S^{(i-1)} \cup \{a^*\}, \frac{\lfloor Mp^* \rfloor}{M}) - f(S^{(i-1)} \cup \{\bar{a}^*\}, \bar{p}^*)}_{\le 0} \le \frac{c_2}{M^2}$$

Merging the two parts, we would have

$$f^*(S^{(i-1)} \cup \{a^*\}) - f^*(S^{(i)}) \le f^*(S^{(i-1)} \cup \{a^*\}) - f(S^{(i)}, p^{(i)}) \le \frac{c_2}{M^2} + 2\sqrt{\frac{2\log(2knMT^2)}{m}}$$

Therefore, by Lemma 2, $f^*(S^{(k)}) + 2k\sqrt{\frac{2\log(2knMT^2)}{m}} + \frac{kc_2}{M^2}$ is greater than $(1 - e^{-(1-\kappa_g)})f^*(S^*)$. Similarly, $f^*(S^{(k)}) - f(S^{(k)}, p^{(k)}) \le \frac{c_2}{M^2} + 2\sqrt{\frac{2\log(2knMT^2)}{m}}$.

Following the proof of Theorem 1, we have

$$\mathbb{E}[R_{1-e^{-(1-\kappa_g)}}] \le 1 + mMnk(1 - e^{-(1-\kappa_g)})f^*(S^*) + \sum_{t \notin \cup_i T_k} 2(k+1)\sqrt{\frac{2\log(2knMT^2)}{m}} + \frac{(k+1)c_2}{M^2}$$

$$\le 1 + mMnk + \frac{T(k+1)c_2}{M^2} + 2T(k+1)\sqrt{\frac{2\log(2knMT^2)}{m}}$$

$$\le 1 + T^{5/7}n^{2/7}k\log^{2/3}(2knT^3) + (k+1)c_2 T^{5/7}n^{2/7}$$

$$+ 2\sqrt{2}(k+1)T^{5/7}\log^{2/3}(2knT^3)$$

where the last lines takes $M = T^{1/7}n^{-1/7}$ and $m = T^{4/7}\log^{1/3}(2knT^3)n^{-4/7}$. $\qquad\square$

## F  GENERAL COST PROFIT MAXIMIZING ALGORITHM

---

**Algorithm 2** Randomized Greedy Dynamic Pricing

---

**Input:** $T$, $m$, $M$ price discretization, $\{c_i\}_1^n$ product costs
**Initialization:** $\bar{P} = \{\frac{1}{M}, \frac{2}{M}, \ldots, \frac{M}{M}\}$, $S^{(0)} = \emptyset$
Add $2k$ dummy products with zero cost and zero utility, and update $n \leftarrow n + 2k$
**for** $i = 1, 2, \ldots, l$ **do**
    **for** $a \notin S^{(i-1)}, \bar{p} \in \bar{P}$ **do**
        **for** $j = 1, \ldots, m$ **do**
            Play $S_a := S^{(i-1)} \cup a$ with price $\bar{p}$, receive reward $r_t = \bar{p}\mathbf{1}(U(S_a) \ge \bar{p})$.
            $t \leftarrow t + 1$
        Compute $\widehat{\mu}_{S_a,\bar{p}} = \frac{1}{m}\sum_{t:(S_a,\bar{p}) \text{ played}} r_t$ for all $a \notin S^{(i-1)}, \bar{p} \in \bar{P}$
    Let $a_i \sim \text{Uniform}(T_i)$ where $T_i = \arg\max_{T \subseteq [n] \setminus S^{(i-1)}, |T|=k} \max_{\bar{p} \in \bar{P}} \sum_{a \in U} (\widehat{\mu}_{S_a,\bar{p}} - c_a)$
    Update the base set: $S^{(i)} \leftarrow S^{(i-1)} \cup \{a_i\}$
    Store the selected price: $p^{(i)} \leftarrow \arg\max_{\bar{p} \in \bar{P}} \widehat{\mu}_{S^{(i)},\bar{p}}$
**while** $t < T$ **do**
    Play $(S^{(k)}, p^{(k)})$ and update $t \leftarrow t + 1$

---

If the cost were added, then the revenue under optimal pricing would be the difference of a monotonic submodular function(Assumption 1) and a modular function (costs of individual products in a bundle), which is submodular but it may be nonmonotonic. The optimal approximation ratio of maximization of nonmonotone submodular functions under cardinality constraint is an open problem, however, Buchbinder et al. (2014) introduced a randomized greedy algorithm capable of $e^{-1}$-approximation of the maximum value. We now generalize this result to show this approximation is robust to local errors.

**Lemma 5.** *(Lemma 2.2 of Buchbinder et al. (2014)) Let $S(p) \subseteq [n]$ be a random subset from a distribution that the marginal probability of every element appearing in the subset is at most $p$ (not necessarily independently ). Then, for every submodular function $g : \mathbf{2}^{[n]} \rightarrow \mathcal{R}$, we have $\mathbb{E}[g(S(p))] \geq (1 - p)g(\emptyset)$*

**Corollary 1.** *For all $0 \leq i \leq k$, $f^*(S^{(i)} \cup S^*) \geq (1 - \frac{1}{k})^i f^*(S^*)$.*

Now we show that the randomized greedy algorithm solution gives an $e^{-1}$- approximation that is is robust to local noise, generalizing Theorem 1.3 of Buchbinder et al. (2014).

**Lemma 6.** *Let $S^{(k)} \supset \cdots \supset S^{(1)}, |S^{(i)}| = i$ be a step-wise **randomized** $\epsilon$-greedy solution for submodular set function $g$, i.e. at each step an element $a_i$ is uniformly selected from subset $T_i \subseteq [n] \backslash S^{(i-1)}$ of size $k$ , where $\max_{T \subseteq [n] \backslash S^{(i-1)}, |T|=k} \sum_{a \in T} g(a) - \sum_{a \in T_i} g(a) \leq k\epsilon$ for all $i$. For any $\epsilon \geq 0$, we have*

$$\mathbb{E}[g(S^{(k)})] + k\epsilon \geq e^{-1}g(S^*).$$

*Proof.* Let $T_i^* = \arg\max_{T \subseteq [n] \backslash S^{(i-1)}, |T|=k} \sum_{a \in T} g(a)$, and $R_i$ be $S^* \backslash S^{(i-1)}$ union enough dummy elements (zero added value to every subset) to make $|R_i| = k$. Now conditional on event with fixed $S^{(i-1)}$,

$$\mathbb{E}[g(S^{(i-1)} \cup \{a_i\}) - g(S^{(i-1)})] = \frac{1}{k} \sum_{b \in T_i} g(S^{(i-1)} \cup \{b\}) - g(S^{(i-1)})$$

$$\geq \frac{1}{k}\Big(\sum_{b \in T_i^*} g(S^{(i-1)} \cup \{b\}) - k\epsilon\Big) - g(S^{(i-1)})$$

$$\geq \frac{1}{k} \sum_{b \in R_i} g(S^{(i-1)} \cup \{b\}) - g(S^{(i-1)}) - \epsilon$$

$$\geq \frac{1}{k}(g(S^* \cup S^{(i-1)}) - g(S^{(i-1)})) - \epsilon$$

Taking expectation over all possible $S^{(i-1)}$, we have

$$\mathbb{E}[g(S^{(i-1)} \cup \{a_i\}) - g(S^{(i-1)})] \geq \frac{1}{k}\Big(\mathbb{E}[g(S^* \cup S^{(i-1)})] - \mathbb{E}[g(S^{(i-1)})]\Big) - \epsilon$$

$$\geq \frac{1}{k}\big(1 - \frac{1}{k}\big)^i g(S^*) - \frac{1}{k}\mathbb{E}[g(S^{(i-1)})] - \epsilon$$

Finally, adding all this inequalities we get,

$$\mathbb{E}[g(S^{(k)})] \geq (1 - \frac{1}{k})^{k-1}g(S^*) - k\epsilon \geq e^{-1}g(S^*) - k\epsilon$$

$\square$

**Theorem 4.** *If Assumption 1 holds, then algorithm 1 with $m = T^{1/2}/\log^{1/2} T$ and $M = T^{1/4}/\log^{1/4} T$ achieves $e^{-1}$-regret $\mathcal{O}(T^{3/4})$.*

*Proof.* Similar to the proof of Theorem 1, with probability of at least $1 - \frac{1}{T}$,

$$\sum_{(\bar{a}^*, \bar{p}^*) \in \bar{T}_i^*} f(S^{(i-1)} \cup \{\bar{a}^*\}, \bar{p}^*) - \sum_{(a, \bar{p}) \in T_i} f(S^{(i-1)} \cup \{a\}, p) \leq 2k\sqrt{\frac{2\log(2knMT^2)}{m}}$$

And

$$\sum_{a^* \in T_i^*} f^*(S^{(i-1)} \cup \{a^*\}) - \sum_{(\bar{a}^*, \bar{p}^*) \in \bar{T}_i^*} f(S^{(i-1)} \cup \{\bar{a}^*\}, \bar{p}^*) \le \frac{k}{M}.$$

Therefore, by [Lemma 6](), $\mathbb{E}[f^*(S^{(k)})] + 2k\sqrt{\frac{2\log(2knMT^2)}{m}} + \frac{k}{M}$ is greater than $e^{-1}f^*(S^*)$.

$$\mathbb{E}[R_{e^{-1}}] \le \mathbb{P}[G^c]T + \mathbb{E}[R_{e^{-1}}\mathbf{1}\{G\}] \le \frac{1}{T}T + \mathbb{E}[R_{e^{-1}}\mathbf{1}\{G\}]$$

$$\le 1 + \sum_{i=1}^{k}\sum_{t \in T_i} e^{-1}f^*(S^*) - \mathbb{E}[f(S^{(i-1)} \cup \{a_t\}))]$$

$$+ \sum_{t \notin \cup_i T_i} e^{-1}f^*(S^*) - \mathbb{E}[f(S^{(k)}, p^{(k)})]$$

$$\le 1 + mMnke^{-1}f^*(S^*) + \sum_{t \notin \cup_i T_k} 2(k+1)\sqrt{\frac{2\log(2knMT^2)}{m}} + \frac{k+1}{M}$$

$$\le 1 + mMnk + \frac{T(k+1)}{M} + 2T(k+1)\sqrt{\frac{2\log(2knMT^2)}{m}}$$

$$\le 1 + T^{3/4}n^{1/4}k\log^{2/3}(2knT^3) + (k+1)T^{3/4}n^{1/4}$$

$$+ 2\sqrt{2}(k+1)T^{3/4}\log^{2/3}(2knT^3)$$

$$\text{(Setting } M = T^{1/4}n^{-1/4} \text{ and } m = T^{1/2}\log^{1/3}(2knT^3)n^{-1/2})$$

$\square$

## G    EXTRA PLOTS/EXAMPLES FOR OFFLINE SETTING

We also investigate bundling products with supermodular valuation to see if bundling supermodular sets increases revenue more than submodular ones. In [Figure 5](), we compare two bundles of size two, bundled in both mixed bundling and pure bundling scenarios. it can be seen that the bundle where the gap between linear price and valuation is negative have an insignificant change in revenue, while the submodular bundle is doubling the revenue.

### G.1    PROOF OF THEOREM 2

Define $f := rev$, and the event $G := \bigcap_{i=1}^{k}\bigcap_{a \in [n]\setminus S^{(i-1)}}\bigcap_{t=1}^{T}\bigcap_{\bar{p} \in \bar{P}} g_{i,a,t,\bar{p}}$ where

$$g_{i,a,t,\bar{p}} := \left\{ \left| \sum_{s \le t: I_s = (S^{(i-1)} \cup \{a\}, \bar{p})} (r_s - f(S^{(i-1)} \cup \{a\}, \bar{p})) \right| \le \sqrt{2T_{(S^{(i-1)} \cup \{a\}, \bar{p})}(t)\log(2knMT^2)} \right\}.$$

where $X_s$ are 1-sub-gassian mean-zero noise as the revenue is bounded in $[0,1]$. then,

$$\mathbb{P}(G^c) \le \sum_{i=1}^{k}\mathbb{P}\Big( \bigcup_{a \in [n]\setminus S^{(i-1)}}\bigcup_{\bar{p} \in \bar{P}}\bigcup_{t=1}^{T} g_{i,a,t,\bar{p}}^c \Big)$$

$$= \sum_{i=1}^{k}\sum_{S \in \binom{[n]}{i-1}}\mathbb{P}\Big( \bigcup_{a \in [n]\setminus S}\bigcup_{\bar{p} \in \bar{P}}\bigcup_{t=1}^{T} g_{i,a,t,\bar{p}}^c | S^{(i-1)} = S \Big)\mathbb{P}(S^{(i-1)} = S)$$

$$\le \sum_{i=1}^{k}\sum_{S \in \binom{[n]}{i-1}}\sum_{a \in [n]\setminus S}\sum_{\bar{p} \in \bar{P}}\mathbb{P}\Big( \bigcup_{t=1}^{T} g_{i,a,t,\bar{p}}^c | S^{(i-1)} = S \Big)\mathbb{P}(S^{(i-1)} = S)$$

$$\le \sum_{i=1}^{k}\sum_{S \in \binom{[n]}{i-1}}\sum_{a \in [n]\setminus S}\sum_{\bar{p} \in \bar{P}}\mathbb{P}\Big( \bigcup_{t=1}^{T}\{| \sum_{s=1}^{t} X_s| \ge \sqrt{2t\log(2knMT^2)}\} \Big)\mathbb{P}(S^{(i-1)} = S)$$

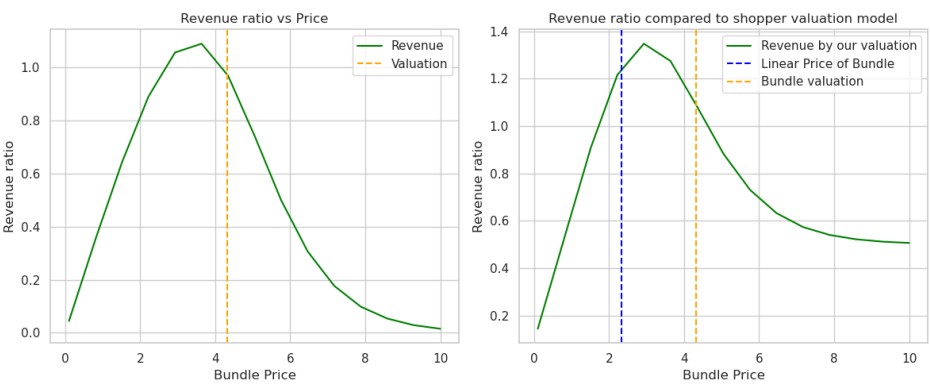

(a) Bundle of two products with supermodular valuation (gap of $0.2028)

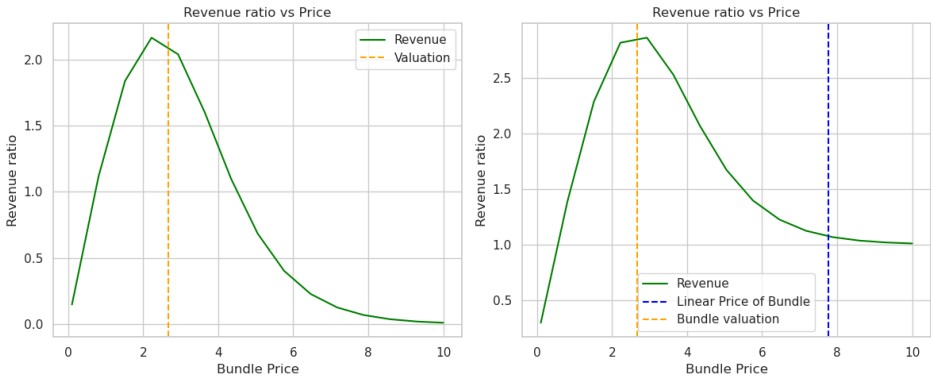

(b) Bundle of two products with submodular valuation (gap of $ - 0.5622$)

Figure 5: Comparing bundles of equal valuation with different gaps of linear price and valuation

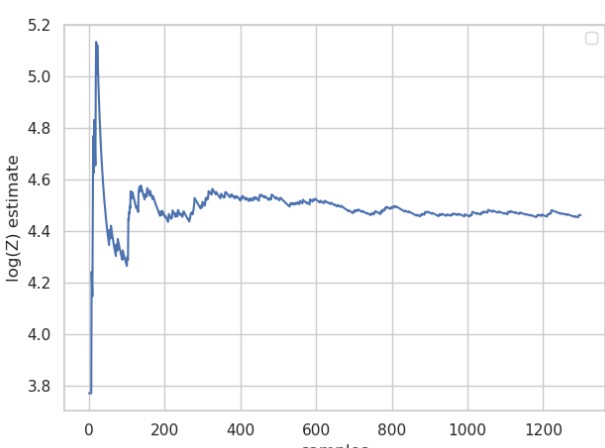

Figure 6: Convergence of $\log Z := \log \sum_S \exp(V(S) - \text{price}(S))$

$$\leq \sum_{i=1}^k \sum_{S \in \binom{[n]}{i-1}} \sum_{a \in [n] \setminus S} \sum_{\bar{p} \in \bar{P}} \sum_{t=1}^T \frac{1}{knMT^2} \mathbb{P}(S^{(i-1)} = S) \leq 1/T.$$

We now prove that on event $G$, the arm and price selected at the $i$-th step of the algorithm is within $2\sqrt{\frac{2\log(2knMT^2)}{m}}$ of the best possible arm+price at that step.

Let $\bar{a}^*, \bar{p}^* = \arg\max_{a \notin S^{(i-1)}, \bar{p} \in \bar{P}} f(S^{(i-1)} \cup \{a\}, \bar{p})$,

$$f(S^{(i-1)} \cup \{\bar{a}^*\}, \bar{p}^*) - f(S^{(i)}, p^{(i)}) \leq f(S^{(i-1)} \cup \{\bar{a}^*\}, \bar{p}^*) - \widehat{\mu}_{S^{(i-1)} \cup \{\bar{a}^*\}, \bar{p}^*}$$
$$+ \underbrace{\widehat{\mu}_{S^{(i-1)} \cup \{\bar{a}^*\}, \bar{p}^*} - \widehat{\mu}_{S^{(i)}, p^{(i)}}}_{\leq 0} + \widehat{\mu}_{S^{(i)}, p^{(i)}} - f(S^{(i)}, p^{(i)})$$
$$\leq 2\sqrt{\frac{2\log(2knMT^2)}{m}}$$

Now let $a^* = \arg\max_a f^*(S^{(i-1)} \cup \{a\})$, where $f^*(S) = \max_{p \in [0,1]} f(S, p)$, and $p^* = \arg\max_{p \in [0,1]} f(S^{(i-1)} \cup a^*, p)$. Noting that $\frac{\lfloor Mp^* \rfloor}{M}$ is in discretized price set $\bar{P}$, we have

$$f(S^{(i-1)} \cup \{a^*\}, p^*) - f(S^{(i-1)} \cup \{\bar{a}^*\}, \bar{p}^*)$$
$$\leq f(S^{(i-1)} \cup \{a^*\}, p^*) - f(S^{(i-1)} \cup \{a^*\}, \frac{\lfloor Mp^* \rfloor}{M})$$
$$+ \underbrace{f(S^{(i-1)} \cup \{a^*\}, \frac{\lfloor Mp^* \rfloor}{M}) - f(S^{(i-1)} \cup \{\bar{a}^*\}, \bar{p}^*)}_{\leq 0}$$
$$\leq (p^* - \frac{\lfloor Mp^* \rfloor}{M}) \mathbb{P}\left[U(S^{(i-1)} \cup \{a^*\}) \geq \frac{\lfloor Mp^* \rfloor}{M}\right] \leq \frac{1}{M}$$

where the last line follows from the facts that the CDF is nondecreasing, and that $(p^* - \frac{\lfloor Mp^* \rfloor}{M}) \leq 1/M$. Merging the two parts, we have

$$f^*(S^{(i-1)} \cup \{a^*\}) - f^*(S^{(i)}) \leq f^*(S^{(i-1)} \cup \{a^*\}) - f(S^{(i)}, p^{(i)}) \leq \frac{1}{M} + 2\sqrt{\frac{2\log(2knMT^2)}{m}}$$

Therefore, $S^{(k)}$ is a $\left(\frac{1}{M} + 2\sqrt{\frac{2\log(2knMT^2)}{m}}\right)$-greedy solution for $f^*$, and by Lemma 2, $f^*(S^{(k)}) +$

$2k\sqrt{\frac{2\log(2knMT^2)}{m}} + \frac{k}{M}$ is greater than $(1 - e^{-(1-\kappa_g)})f^*(S^*)$. Moreover,

$$f^*(S^{(k)}) - f(S^{(k)}, p^{(k)}) \le f^*(S^{(k-1)} \cup \{a^*\}) - f(S^{(k)}, p^{(k)}) \le \frac{1}{M} + 2\sqrt{\frac{2\log(2knMT^2)}{m}}$$

Let $T_i$ be the set of times where we pulled a set of cardinality $i$ before committing to a set and price, so we have $|T_i| \le mMn$ for $i \le k$. Therefore,

$$\mathbb{E}[R_{1-e^{-(1-\kappa_g)}}] \le \mathbb{P}[G^c]T + \mathbb{E}[R_{1-e^{-(1-\kappa_g)}}\mathbf{1}\{G\}] \le \frac{1}{T}T + \mathbb{E}[R_{1-e^{-(1-\kappa_g)}}\mathbf{1}\{G\}]$$

$$\le 1 + \sum_{i=1}^{k}\sum_{t\in T_i}(1 - e^{-(1-\kappa_g)})f^*(S^*) - f(S^{(i-1)} \cup \{a_t\}))$$

$$+ \sum_{t\notin\cup_i T_i}(1 - e^{-(1-\kappa_g)})f^*(S^*) - f(S^{(k)}, p^{(k)})$$

$$\le 1 + mMnk(1 - e^{-(1-\kappa_g)})f^*(S^*) + \sum_{t\notin\cup_i T_i}2(k+1)\sqrt{\frac{2\log(2knMT^2)}{m}} + \frac{k+1}{M}$$

$$\le 1 + mMnk + \frac{T(k+1)}{M} + 2T(k+1)\sqrt{\frac{2\log(2knMT^2)}{m}}$$

$$\le 1 + T^{3/4}n^{1/4}k\log^{2/3}(2knT^3) + (k+1)T^{3/4}n^{1/4} + 2\sqrt{2}(k+1)T^{3/4}\log^{2/3}(2knT^3)$$

where the last line follows from taking $M = T^{1/4}n^{-1/4}$ and $m = T^{1/2}\log^{1/3}(2knT^3)n^{-1/2}$.

## H ALGORITHM SIMULATION

We use the model trained on the cosmetics dataset in Section 2 as an oracle for the valuation of each set of products, employing an exponential demand function with $\lambda = \frac{1}{V(S)}$ at each time $t$ to satisfy Assumption 1. Note that our quadratic model of valuation in the offline setting can be decomposed into the sum of a submodular function (all diagonal and negative off-diagonal values of matrix $A$) and a supermodular function (all non-negative off-diagonal values of $A$). As the exact supermodular curvature of the revenue function is hard to compute, we plot the regret against the offline greedy solution, which serves as an upper bound of $\alpha$-regret.

We observe in Figure 7 that as $T$ increases, the algorithm commits to a better greedy set, and the regret approaches a slope of $3/4$. The jumps in regret occur because the discretization level is increasing. Note that the plot is double logarithmic.

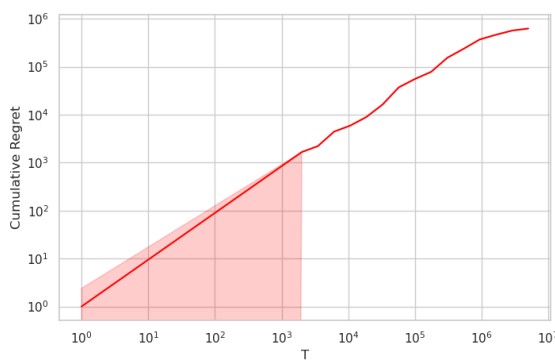

Figure 7: The performance of Algorithm 1 on the cosmetics dataset, where regret is measured against the offline greedy solution with $k = 4$.

