# OpenReview forum: "Optimal Pricing for Bundles: Using Submodularity in Offline and Online Settings"
_ICLR.cc/2026/Conference — ICLR 2026 Conference Withdrawn Submission_

### Official Review · Reviewer_ncsN · 2025-10-28

**Soundness:** 1
**Presentation:** 2
**Contribution:** 2
**Rating:** 2
**Confidence:** 4

**Summary:**

This paper presents a framework for identifying revenue-maximizing product bundles and their optimal prices, exploring both offline and online settings. The core idea is that submodularity is an effective and sample-efficient criterion for finding promising bundles.

**Strengths:**

**Originality & Significance:** The paper's primary contribution is framing the problem of revenue-maximizing bundle pricing. It studies this problem under a submodularity assumption. The work focuses on both offline and online settings. It provides an online learning algorithm for the online setting, while also conducting experiments in a data-rich offline setting.

**Quality:** In the offline setting, the choice of a quadratic valuation model is well-justified as it has a direct connection to the submodularity of product pairs. The methodology for fitting this model and then using it to estimate the revenue impact of new bundles is sound and clearly explained. In the online setting, the authors use reasonable sets of assumptions (Assumptions 1 and 2) to develop a greedy algorithm (Algorithm 1) and provide a regret analysis.

**Clarity:** The introduction provides good motivation, using relatable examples and building intuition for the core concept of submodularity. The distinction between the offline and online problems is sharp.

**Weaknesses:**

**Missing Proofs:** This is the most critical weakness. The paper claims that proofs for Theorem 1 and Lemma 2 are in Appendix D. However, Appendix D only contains a proof for a different result (Lemma 3). The core theoretical results of the online section are therefore unsubstantiated.

**Lack of Formalism and Algorithmic Detail in Offline Setting:** The exposition of the offline setting lacks rigor.
- It focuses heavily on a motivating example rather than formally defining the revenue maximization problem and presenting a general algorithm to solve it.
- Lemma 1 connects the submodularity gap to revenue increase for pairs of products. The authors claim this "facilitates the use of efficient approximation algorithms," but never describe such an algorithm for the general case of finding the best bundle of size $k$.

**Technical Novelty in Online Algorithm:** Algorithm 1 is an "explore-then-commit" greedy algorithm. This is a well-established paradigm in the literature on online submodular maximization. While its application to the bundle pricing problem is novel, the paper could do more to delineate the specific technical innovations in the regret analysis compared to prior work. The key challenge here is jointly optimizing over the combinatorial set of bundles and the continuous set of prices. The paper handles this via discretization, but a more detailed discussion on the unique challenges posed by the pricing dimension and how the analysis addresses them would better highlight the technical contribution.

**Structural and Presentation Issues:** The paper's structure and presentation could be significantly improved.
- The online setting contains the paper's most substantial technical results and should be presented first. The current ordering buries the lead behind a less formal and less complete offline analysis.
- The lengthy motivating example in Section 2 (as well as Figures 2 and 3) interrupts the paper's flow. This discussion would be better suited as a case study in the appendix, allowing the main body to focus on the core technical contributions.
- The quality of the figures is low. Figure 2, which plots demand curves, is not very informative as it primarily shows that demand for individual items decreases as a competing bundle's price drops, which is an expected outcome. The figures need to be better designed to convey the key insights more effectively.
- Furthermore, the appendix is disorganized. For example, Appendix G, titled "EXTRA PLOTS/EXAMPLES FOR OFFLINE SETTING," contains the proof for Theorem 2, which is interrupted by plots, making it difficult to follow. The appendix must be thoroughly reorganized and, most importantly, the missing proofs must be included.

**Questions:**

- The proofs for Theorem 1 and Lemma 2, which are the main theoretical results for the online setting, appear to be missing from the manuscript. The paper states that it is in Appendix D, but that section contains a proof for a different result. Could you please provide these proofs or clarify their location?
- The analysis in the offline setting focuses heavily on a motivating example. Could you please provide a more formal problem definition for revenue maximization, given the learned quadratic valuation function?
- You state that Lemma 1, which applies to bundles of size $k=2$, "facilitates the use of efficient approximation algorithms" for finding promising bundles in general. How does one use this property to search the combinatorial space of bundles efficiently? The paper does not provide an answer beyond the simple case study.

---

### Official Review · Reviewer_hqZr · 2025-10-30

**Soundness:** 3
**Presentation:** 3
**Contribution:** 3
**Rating:** 4
**Confidence:** 3

**Summary:**

This submission studies revenue-maximizing bundle pricing under a cardinality constraint for customers with unknown valuation functions. Two variants are presented: offline (learning from historical baskets, modeled with a logit choice model) and online (sequential interaction with sale/no-sale feedback). The goal is to identify promising bundles and their prices in the offline case, and to learn a near-optimal bundle–price pair with regret guarantees in the online case.

More specifically, in the offline setting one is provided with prices of the single items and a list of bundles that have been bought by the customers. Then the goal is to learn from this data optimal prices for bundles. To make that feasible, a specific class of valuation functions is introduced that captures both submodular and supermodular valuations. This class has $n^2$ parameters if there are n items (one for each pair of items modelling their joint value). Then a method for learning these parameters based on gradient descent is provided and testet on a real-world data set.

For the online setting, the submission proposes an explore-then-commit greedy approach that builds a bundle of size k greedily by estimating marginal revenue gains for candidate items and discretizes prices to search over a grid. Under typical assumptions on the pricing function, a regret bound of O ⁣(T^3/4n^1/4k) against an α-approximation to the hindsight optimum is proven. When the demand curve is concave in price, the regret improves to the order of T^5/7.

**Strengths:**

Most of the proofs are technically sound and well written. In the online setting an algorithm with a provable regret bound is obtained.

**Weaknesses:**

The part about the offline setting does not contain any particular interesting or new results. A natural model is introduced and a standard framework to learn its parameters is used. The algorithm for the online setting is also quite natural: it starts with an exploration phase, in which a greedy algorithm is applied to obtain a good bundle-price combination. Then the submodularity guarantees that the greedy algorithm finds a good solution. I find the online model also somewhat questionable because each buyer is presented only with a single bundle that she can take or leave. In the motivating examples in the paper, a model in which multiple bundles are offered and one can also buy single items separately seems more realistic.

**Questions:**

Remark 2, which claims that the demand function $Pr[U(S) \geq p]$ is continuous if and only if $rev(p, S)$ is 1-Lipschitz, is likely incorrect or, at least, requires further justification. This property is used in the greedy algorithm's analysis to provide an upper bound for the error introduced by price discretization. For this purpose, it would be sufficient to show $L$-Lipschitz continuity for some finite $L > 0$, which seems more plausible.

---

### Official Review · Reviewer_Jbah · 2025-10-30

**Soundness:** 2
**Presentation:** 2
**Contribution:** 3
**Rating:** 4
**Confidence:** 3

**Summary:**

The paper looks at bundle pricing in two scenarios - offline where you have historical purchase data, and online where customers arrive sequentially. The main idea is that products with "submodular" valuations (where the bundle is worth less than sum of parts) are actually good candidates for bundling because you can discount them and drive demand. They fit a quadratic model to retail data and show correlation between submodularity and revenue gains. For online they give an algorithm with T^3/4 regret.

**Strengths:**

The submodularity angle is interesting and a bit counter-intuitive. I liked Proposition 1 even though it's only stated informally at first (more on this later...)

Real data from an actual retailer is nice to see, even if the dataset is limited

The theoretical result for online setting seems technically sound

Figure 1 makes the main point pretty clearly

**Weaknesses:**

My main concern is that the offline and online parts feel like two different papers. The offline uses a specific logit model with quadratic valuations, then the online suddenly switches to a completely general nonparametric model. Why? If the quadratic model works offline, why not use it online? If the general model is better, why bother with the quadratic one? The paper never really addresses this disconnect.
The experimental validation is pretty weak honestly. For offline, they only look at beauty products from one store, and the baskets are tiny (average 1.19 products - see Fig 4). That's basically people buying 1 item most of the time. How do you even validate bundling when people rarely buy multiple items? For online there's only simulations using the fitted model, no real data. Would have liked to see comparisons to baselines too - how does this compare to just using UCB on bundles, or Thompson sampling, or even random exploration?

The quadratic model seems pretty limiting. It's only pairwise interactions right? What if there are three-way or higher order effects? Like maybe shampoo + conditioner is fine, and shampoo + soap is fine, but shampoo + conditioner + soap together is redundant? The model can't capture that. The paper mentions this has 2^n degrees of freedom without structure but then the quadratic model has n^2 parameters which is still a lot when n=282 products and you only have 2148 receipts. That's a lot of parameters to fit with limited data...

Theoretical issues: The α-regret thing is confusing. The paper says α ≤ 1-e^(-1) but then later it depends on κ_g which is never computed for any real application. So we dont really know if we're getting 0.63-regret or 0.1-regret or what. This makes it hard to evaluate if the bounds are meaningful. Also Assumption 1 about BP decomposition seems strong - when does revenue actually decompose this way? The paper just asserts it. When doesn't it compose?

Proposition 1 / Lemma 1 mismatch is weird. In the intro it's stated generally but then the actual lemma only works for k=2 and requires V({x}) = V({y}). That's much more restrictive.

Presentation: Some parts are hard to follow. The connection between sections 2 and 3 is abrupt. Also the paper introduces this elaborate importance sampling scheme in Appendix B.1 to handle the intractable denominator but doesn't really justify why this is the right approach vs other approximations. Also - no Appendix A?

The pure bundling analysis (Figure 3b) is interesting - shows mixed bundling is better - but this isn't really developed. Seems like an important practical insight that gets buried.

Missing related work: What about the assortment optimization literature? That seems very related. Also recent contextual bandit work with deep learning could be applicable here.

**Questions:**

Why not use the quadratic model in online setting?

What's a typical value of κ_g? Is 0.5 realistic? 0.9?

The dataset is very sparse - did you try any regularization besides ridge with λ=0.01?

How does the explore phase scale - for n=1000, k=10, M=100, m=1000 that's like 10^7 rounds before you commit? Is that correct?

---

### Official Review · Reviewer_HMYJ · 2025-10-31

**Soundness:** 2
**Presentation:** 2
**Contribution:** 2
**Rating:** 6
**Confidence:** 2

**Summary:**

This paper studies the problem of revenue-maximizing bundle pricing under a cardinality constraint, where the seller can choose up to k items to form a bundle and set a price, while buyers’ valuations are unknown and their purchasing decisions follow a choice model based on surplus value. The analysis is conducted under two data settings. In the offline setting, using historical data and assuming a Logit choice model, the authors identify near-optimal bundles and show that the submodularity of the bundle valuation function serves as an efficient criterion for selection. In the online setting, the paper proposes an algorithm to find the optimal bundle–price combination that maximizes revenue, achieving a regret of $T^{3/4}$ against an α-approximation of the optimal revenue.

**Strengths:**

The paper analyzes revenue-maximizing bundle pricing under both offline and online settings. In the offline setting, it leverages historical data and a logit choice model to estimate customer valuations and efficiently identify promising bundles based on submodularity principles. In the online setting, where valuations are unknown and only sale feedback is available, the authors propose an algorithm with a $T^{3/4}$ regret bound. The paper not only provides rigorous theoretical guarantees but also validates the effectiveness of the proposed algorithm through simulation experiments based on real-world data models.

**Weaknesses:**

1. In the offline setting, the valuation function $V(S)$ is defined as a unified function shared by all customers. However, this approach overlooks the significant differences in customer preferences in real-world scenarios, which may lead to results that deviate from practical outcomes.

2. In the online setting, the paper does not compare its proposed algorithm with other similar approaches. As a result, the experimental validation is somewhat limited in its scope.

**Questions:**

In the offline setting, the paper mentions how to identify the most promising bundling combinations, but lacks detailed explanation on the process for determining the optimal pricing. Could the authors provide further clarification on the specific implementation or approach used for this?

---

### Note · Authors · 2025-11-17

I have read and agree with the venue's withdrawal policy on behalf of myself and my co-authors.